# Compact and highly active next-generation libraries for CRISPR-mediated gene repression and activation

Max A Horlbeck[1,2,3,4], Luke A Gilbert[1,2,3,4], Jacqueline E Villalta[1,2,3,4†], Britt Adamson[1,2,3,4], Ryan A Pak[1,5], Yuwen Chen[1,2,3,4], Alexander P Fields[1,2,3,4], Chong Yon Park[1,5], Jacob E Corn[5,6], Martin Kampmann[1,2,3,4,7], Jonathan S Weissman[1,2,3,4*]

[1]Department of Cellular and Molecular Pharmacology, University of California, San Francisco, San Francisco, United States; [2]Howard Hughes Medical Institute, University of California, San Francisco, San Francisco, United States; [3]California Institute for Quantitative Biomedical Research, University of California, San Francisco, San Francisco, United States; [4]Center for RNA Systems Biology, University of California, San Francisco, San Francisco, United States; [5]Innovative Genomics Initiative, University of California, Berkeley, Berkeley, United States; [6]Department of Molecular and Cell Biology, University of California, Berkeley, Berkeley, United States; [7]Institute for Neurodegenerative Diseases, University of California, San Francisco, San Francisco, United states

*For correspondence: Jonathan. Weissman@ucsf.edu

Present address: †Calico Life Sciences LLC, South San Francisco, United States

**Abstract** We recently found that nucleosomes directly block access of CRISPR/Cas9 to DNA (*Horlbeck et al., 2016*). Here, we build on this observation with a comprehensive algorithm that incorporates chromatin, position, and sequence features to accurately predict highly effective single guide RNAs (sgRNAs) for targeting nuclease-dead Cas9-mediated transcriptional repression (CRISPRi) and activation (CRISPRa). We use this algorithm to design next-generation genome-scale CRISPRi and CRISPRa libraries targeting human and mouse genomes. A CRISPRi screen for essential genes in K562 cells demonstrates that the large majority of sgRNAs are highly active. We also find CRISPRi does not exhibit any detectable non-specific toxicity recently observed with CRISPR nuclease approaches. Precision-recall analysis shows that we detect over 90% of essential genes with minimal false positives using a compact 5 sgRNA/gene library. Our results establish CRISPRi and CRISPRa as premier tools for loss- or gain-of-function studies and provide a general strategy for identifying Cas9 target sites.

## Introduction

Highly multiplexed pooled genetic screening methodologies have emerged as powerful and broadly accessible tools for systematically profiling gene function at the scale of mammalian genomes (*Paddison et al., 2004*). Recently, a number of pooled screening platforms have been developed that utilize the bacterial CRISPR (Clustered Regularly Interspaced Palindromic Repeats)-associated nuclease Cas9 paired with libraries of single-guide RNAs (sgRNAs) to disrupt targeted genes (reviewed in *Shalem et al., 2015*). We and others have developed tools based on nuclease-dead Cas9 (dCas9) (*Qi et al., 2013*) to programmably interfere with (CRISPRi) or activate (CRISPRa) transcription (*Gilbert et al., 2014*, *2013*; *Konermann et al., 2015*; *Maeder et al., 2013*; *Perez-Pinera et al., 2013*; *Tanenbaum et al., 2014*), and used these to systematically manipulate gene

expression at genome scale (*Gilbert et al., 2014*; *Konermann et al., 2015*). Together, these screening platforms represent a powerful toolkit for unbiased forward gain-of-function and loss-of-function genetic screens in mammalian cells.

A key step in implementing CRISPR genetic screens is selecting sgRNAs that mediate high Cas9 activity. We and others recently found that nucleosomes provide a direct and profound impediment to Cas9 access to DNA (*Hinz et al., 2015*; *Horlbeck et al., 2016*; *Isaac et al., 2016*), an observation we expected to be particularly important for applications such as CRISPRi and CRISPRa, which require sustained binding of dCas9 to DNA. We found that nucleosome occupancy was predictive of Cas9 activity complementary to and independent of previously described sgRNA sequence features (*Chari et al., 2015*; *Doench et al., 2014*; *Xu et al., 2015*), adding an additional dimension to the set of parameters expected to influence Cas9 activity. These observations, along with the strong nucleosome-dependent phasing observed downstream of the FANTOM consortium-annotated transcription start site (TSS) (*FANTOM Consortium and the RIKEN PMI and CLST (DGT) et al., 2014*; *Horlbeck et al., 2016*), suggested that a quantitative model incorporating all of these features could greatly enhance our ability to predict highly active sgRNAs for CRISPRi and CRISPRa.

To test this, we developed a comprehensive machine learning pipeline trained on data collected from 30 CRISPRi and 9 CRISPRa screens. We found that the resulting models were highly predictive of sgRNA efficacy and strongly weighted nucleosome positioning and specific sequence features. We used these models to design and generate CRISPRi and CRISPRa version 2 (v2) libraries, targeting human and mouse genomes, which are greatly enriched for sgRNAs with high predicted activity. These libraries include several additional improvements, including the option to screen with either 10 sgRNAs per gene or a compact half-library containing the top 5 predicted sgRNAs for each gene. To benchmark this new algorithm, we validated the human CRISPRi v2 (hCRISPRi-v2) library with a screen designed to identify genes essential for robust cell growth. In this experiment, essential genes represent a large class of expected positive controls. We identified over 2100 essential genes with high statistical confidence, significantly improving upon our CRISPRi v1 library (*Gilbert et al., 2014*), and precision-recall analysis showed increased discrimination of gold standard essential and non-essential genes with both 10 sgRNA/gene and 5 sgRNA/gene hCRISPRi-v2 libraries (*Hart et al., 2014*). A large majority of the hCRISPRi-v2 sgRNAs targeting known essential genes produced robust growth phenotypes, a key advance over previous CRISPRi libraries (*Evers et al., 2016*), and our algorithm can accurately predict sgRNA activity for data from screens performed with an independently designed library and in different cell types. Furthermore, we observed that CRISPRi lacked any detectable non-specific toxicity associated with genomic DNA breaks and repair, enabling sensitive detection of genes with subtle growth phenotypes. We also conducted a screen for genes that modify growth rates upon overexpression with our hCRISPRa-v2 and found this identified 60% more genes, with greater enrichment for previously established classes of hit genes, than our version 1 CRISPRa screen. Our results suggest that the CRISPRi and CRISPRa v2 libraries have numerous favorable properties relative to alternate approaches as a resource for targeted or genome-scale loss-of-function and gain-of-function studies in mammalian cells.

## Results

### An integrated machine learning approach predicts highly active sgRNAs for CRISPRi and CRISPRa

We sought to improve upon our first generation CRISPRi and CRISPRa libraries by taking a comprehensive approach that incorporated nucleosome positioning, sequence features, refinement of our original sgRNA design rules, and other potentially informative factors. In order to quantitatively model the contribution of these features to CRISPRi activity, we turned to our recently described CRISPRi activity dataset (*Horlbeck et al., 2016*) in which we integrated data from 30 CRISPRi screens to select 1,539 high-confidence hit genes, and normalized the phenotypes for sgRNAs targeting each gene to the strongest sgRNAs for that gene, resulting in 'activity scores' for 18,380 sgRNAs. We used this set as training data for elastic net linear regression (*Figure 1A*) (*Hui Zou, 2005*). As many of the features included in the model were nonlinear with activity, we first adapted each feature set according to its relationship with activity. Categorical and non-linear parameters were binned prior to linear regression. Because the relationship between CRISPRi activity and target

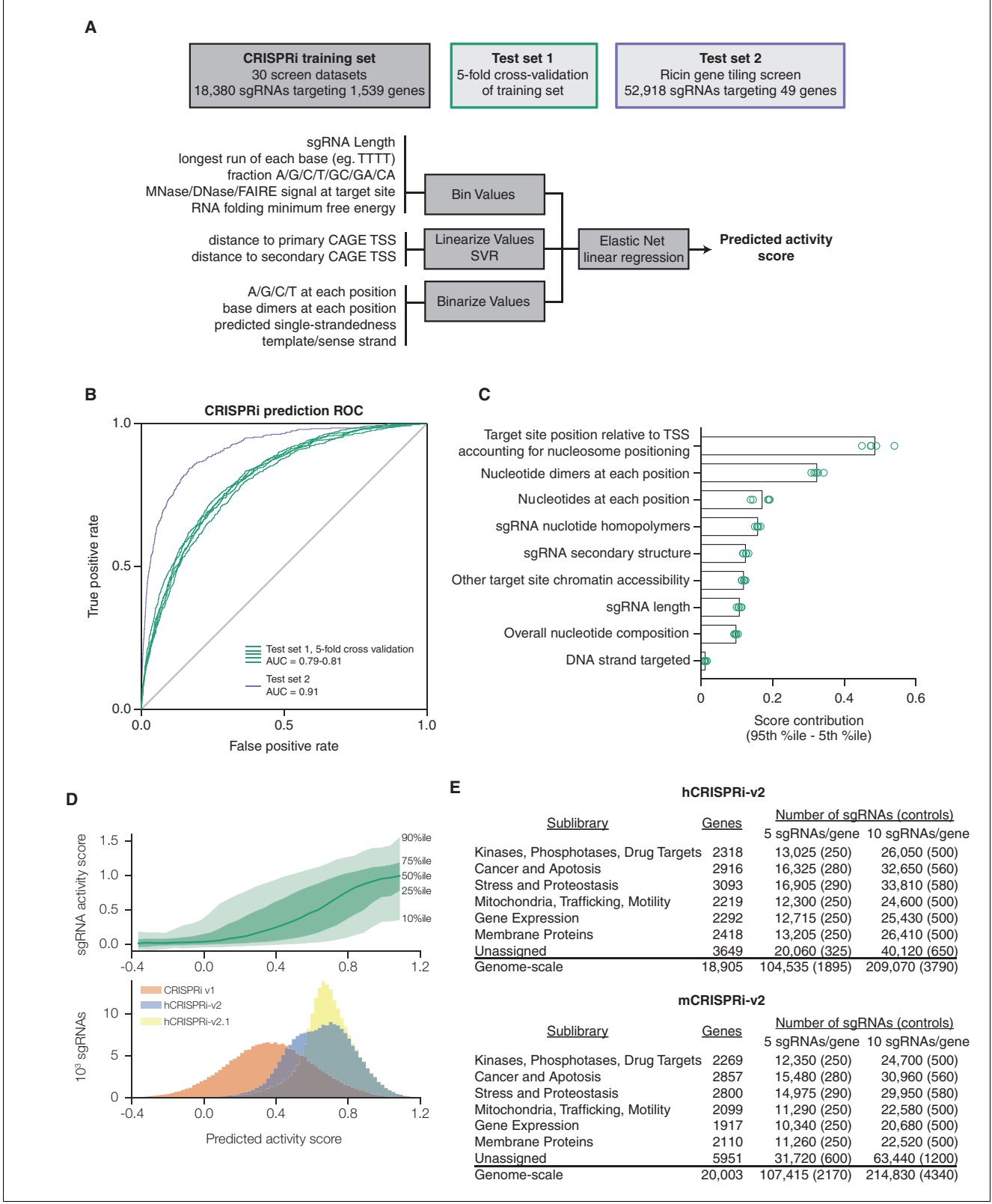

**Figure 1.** A machine learning approach for identifying highly active sgRNAs for CRISPRi. (**A**) Schematic of machine learning strategy and datasets. 808 features were calculated for each sgRNA, linearized as indicated, and z-standardized. A linear regression model was then generated using these features to fit to the activity scores of the CRISPRi training set (*Horlbeck et al., 2016*). 20% of the genes in the training set were reserved to test the predictive value of the resulting model. For display in *Figures 1B,C*, and *Figure 1—figure supplement 2*, five non-overlapping 20% datasets were

*Figure 1 continued*

randomly selected and training was performed on the corresponding 80% sets. An orthogonal dataset, based on tiling of every possible sgRNA within 10 kb of the TSS of 49 genes known to modulate sensitivity to ricin (*Bassik et al., 2013*; *Gilbert et al., 2014*), was also used to assess the predictive value of this model. (B) ROC analysis of the ability of the machine learning approach in (A) to predict highly active sgRNAs. For test set 1, sgRNAs with an activity score greater than 0.75 were considered highly active. For test set 2, sgRNAs with a phenotype greater than 0.75 of the maximum phenotype for each gene were considered highly active. (C) Relative contribution of feature categories to the sgRNA predicted scores. The individual weighting of each feature assigned by the linear regression model (see *Figure 2—figure supplement 2*) was grouped by the indicated categories, and the summed weights for each sgRNA within the 20% test datasets was calculated. The scores of the 95th and 5th percentile sgRNAs were subtracted to compute the overall contribution of the feature category to the distribution of predicted activity scores. Bars indicate the mean of the contributions from five 20% datasets (green circles). The target site position includes both the distance to the TSS and the periodic relationship as fit by SVR (*Figure 1—figure supplement 1*). (D) Distribution of predicted activity scores in next-generation CRISPRi libraries. (Top) Predicted CRISPRi activity correlates with empirical activity scores. For the 80%/20% division used to predict sgRNAs for the hCRISPRi-v2.1 library, predicted scores for the 20% test set were plotted against the empirical activity score. Activity score percentiles are from all sgRNAs within 0.25 of the indicated activity score. Predicted activity was highly correlated with activity, with a Pearson R of 0.56 (p<10$^{-296}$). (Bottom) Distribution of predicted activity scores for CRISPRi v1, hCRISPRi-v2, and hCRISPRi-v2.1, as calculated by the hCRISPRi-v2.1 regression model. (E) Composition of hCRISPRi-v2 and mCRISPRi-v2 sublibraries.

The following figure supplements are available for figure 1:

**Figure supplement 1.** Relationship between CRISPRi activity and sgRNA position relative to the TSS as predicted by SVR.

**Figure supplement 2.** Individual sgRNA feature contributions to predicted CRISPRi activity.

site distance from the TSS was highly periodic and asymmetric, as we had recently shown (*Horlbeck et al., 2016*), we fit sgRNA positioning features using support vector regression (SVR) to predict a continuous function for any target site (Supplementary *Figure 1*). An important improvement was the use of FANTOM consortium annotations instead of Ensembl/GENCODE to define the TSS (*Cunningham et al., 2015*; *FANTOM Consortium and the RIKEN PMI and CLST (DGT) et al., 2014*; *Harrow et al., 2012*) (*Supplementary file 2*), a finding also recently reported by Radzisheus-kaya and colleagues (*Radzisheuskaya et al., 2016*).

We first evaluated the performance of this algorithm using five-fold cross-validation. By performing regression on a training set of only 80% of the genes in the CRISPRi activity dataset, we found that the model was highly predictive of activity in the test set comprising the remaining 20% of the genes, with a receiver operating characteristic area under the curve (ROC-AUC) of 0.80 (*Figure 1B*). Importantly, this high predictive value was consistent across randomly selected training and test sets. As the sgRNAs in this dataset were pre-selected using our CRISPRi v1 design rules, we also tested our ability to predict the performance of sgRNAs in our previously published data set in which we tiled every target site around the TSS of 49 genes known to modulate resistance or sensitivity to the toxin ricin (*Gilbert et al., 2014*) and obtained an ROC-AUC of 0.91 (*Figure 1B*), indicating that we could identify active sgRNAs in the genome with high accuracy.

We next analyzed which features contributed most to CRISPRi activity in the predictive model (*Figure 1—figure supplement 2*). Overall, the predicted scores were most influenced by the position relative to the TSS, including both distance from the TSS and avoidance of canonical nucleosome-occupied regions (*Figure 1C* and *Figure 1—figure supplement 1*). In particular, the nucleosome-deprived region immediately downstream of the TSS yielded the strongest predicted activity by SVR, likely due in part to the contribution of dCas9 directly interfering with early transcriptional initiation or elongation (*Gilbert et al., 2013*; *Qi et al., 2013*). Sequence features also represented a large contribution to the model, and salient relationships included the disfavoring of guanine directly downstream of the protospacer adjacent motif (PAM) which recapitulated previous findings (*Doench et al., 2014*; *Xu et al., 2015*). Additional parameters that contributed to the prediction, included sgRNA secondary structure as predicted by ViennaRNA (*Doench et al., 2016*; *Lorenz et al., 2011*), sgRNA protospacer length, and chromatin accessibility features not accounted for by the nucleosome positioning relationship. The contribution of each individual parameter was also remarkably consistent across 80%/20% divisions of the training dataset, suggesting that the model was capturing underlying biological signal rather than overfitting the data.

Having established the robustness and accuracy of this approach, we used a version of our sgRNA predictions to design a CRISPRi genome-scale library targeting the human protein-coding

transcriptome (hCRISPRi-v2) (*Supplementary file 3*; an *in silico* library design based on the final version of our predictions, hCRISPRi-v2.1, is also available in *Supplementary file 3*). While the predicted scores for sgRNAs in our v1 library were broadly distributed, many sgRNAs of higher predicted activity were available in the genome, and by picking the top 10 predicted sgRNAs per gene we expected that we could greatly enrich the library for guides of high activity (*Figure 1D*). In constructing this library, we also incorporated empirical information for highly active sgRNAs where available, revised off-target filtering, and implemented changes to the sgRNA expression vector to facilitate the processing of screen samples for sequencing (see Materials and methods). In addition, we cloned the library as separate pools for the top 5 and next-best 5 predicted sgRNAs per gene to facilitate screens where a smaller library may be advantageous, and further divided the pools into 7 thematic sublibraries based on our previous divisions of shRNA libraries (*Kampmann et al., 2015*) (*Figure 1E*).

We then used the same approach to design a next-generation CRISPRa library. Due to the requirement for CRISPRa targeting to be upstream of the TSS for maximal activity (*Gilbert et al., 2014*), we collected an activity score dataset of 2,898 sgRNAs from 9 CRISPRa screens (Y.C. and M. K., personal communication) to train an independent predictive model (*Figure 2A*). While the input set was significantly smaller than for CRISPRi, the resulting linear regression still had good predictive value (ROC-AUC 0.70; *Figure 2B*) and generally shared features of the CRISPRi model (*Figure 2C* and *Figure 2—figure supplement 1–2*). We observed that periodic relationship between distance to the TSS and sgRNA activity was less pronounced for CRISPRa than for CRISPRi (*Figure 2—figure supplement 1*; *Figure 1—figure supplement 1*), a difference we attributed to the reduced dynamic range in the nucleosome-depleted region around the TSS, and the smaller number and relatively lower expression of the genes targeted in the CRISPRa training dataset. We used the top 10 predicted sgRNAs for each gene to construct a next-generation library, which significantly increased the predicted activity of the library over our v1 designs (*Figure 2D*). The hCRISPRa-v2 library was partitioned into 14 sublibraries as described for the v2 CRISPRi library above (*Figure 2E*). Importantly, while several strategies have been described for CRISPR-mediated activation (*Chavez et al., 2015*; *Gilbert et al., 2013*; *Hilton et al., 2015*; *Konermann et al., 2015*; *Maeder et al., 2013*; *Perez-Pinera et al., 2013*; *Tanenbaum et al., 2014*; *Zalatan et al., 2015*), a recent comparison of these strategies observed that sgRNA activity generally correlated across all approaches (*Chavez et al., 2016*). Our CRISPRa-v2 libraries are thus likely to serve as a valuable resource for effectively targeting most activator systems.

Finally, we applied the CRISPRi and CRISPRa models to predict highly active sgRNAs targeting the mouse protein-coding transcriptome and generated corresponding genome-scale libraries (mCRISRPi-v2 and mCRISPRa-v2) (*Figures 1E*, *2E*). All four library designs are included as *Supplementary file 3–6*, sgRNA prediction and library design scripts are available online (see Materials and methods), and the cloned lentiviral libraries are available on Addgene.

## The large majority of sgRNAs in the hCRISPRi-v2 library are effective

A central goal in developing the hCRISPRi-v2 library was to enrich for highly active sgRNAs, which would improve statistical confidence in hits and enable high sensitivity even with a compact 5 sgRNA/gene library. Underscoring the importance of this goal, Evers et al. generated a small-scale CRISPRi library targeting predetermined sets of essential and non-essential genes and conducted screens for growth phenotypes in the bladder carcinoma cell line RT-112, and found that ~50% of the CRISPRi sgRNAs targeting essential genes were inactive (*Evers et al., 2016*), similar to rates observed using our CRISPRi v1 library. To test whether our next-generation sgRNA prediction algorithms were able to identify these inactive sgRNAs, we evaluated the predicted activity of the sgRNAs targeting known essential genes in this screen. Despite the screen being performed with an independently designed library, using an sgRNA constant region we have found to underperform our current design (*Chen et al., 2013*), and in a cell type not evaluated in any of our training or test datasets, predicted activity correlated well with the sgRNA growth phenotypes observed in the screen (*Figure 3A*; Pearson R = −0.58, p<10$^{-37}$). Over 40% of the sgRNAs in this library had a predicted activity score less than 0.4, a regime in which the vast majority (over 87%) of sgRNAs were inactive, as defined by z-score > −2 relative to non-essential gene-targeting sgRNAs, enabling *a priori* elimination of 60% of the inactive sgRNAs by simply applying this threshold. By contrast, of the sgRNAs in that library with predicted activity scores ≥0.6, 77% are active. Use of the improved

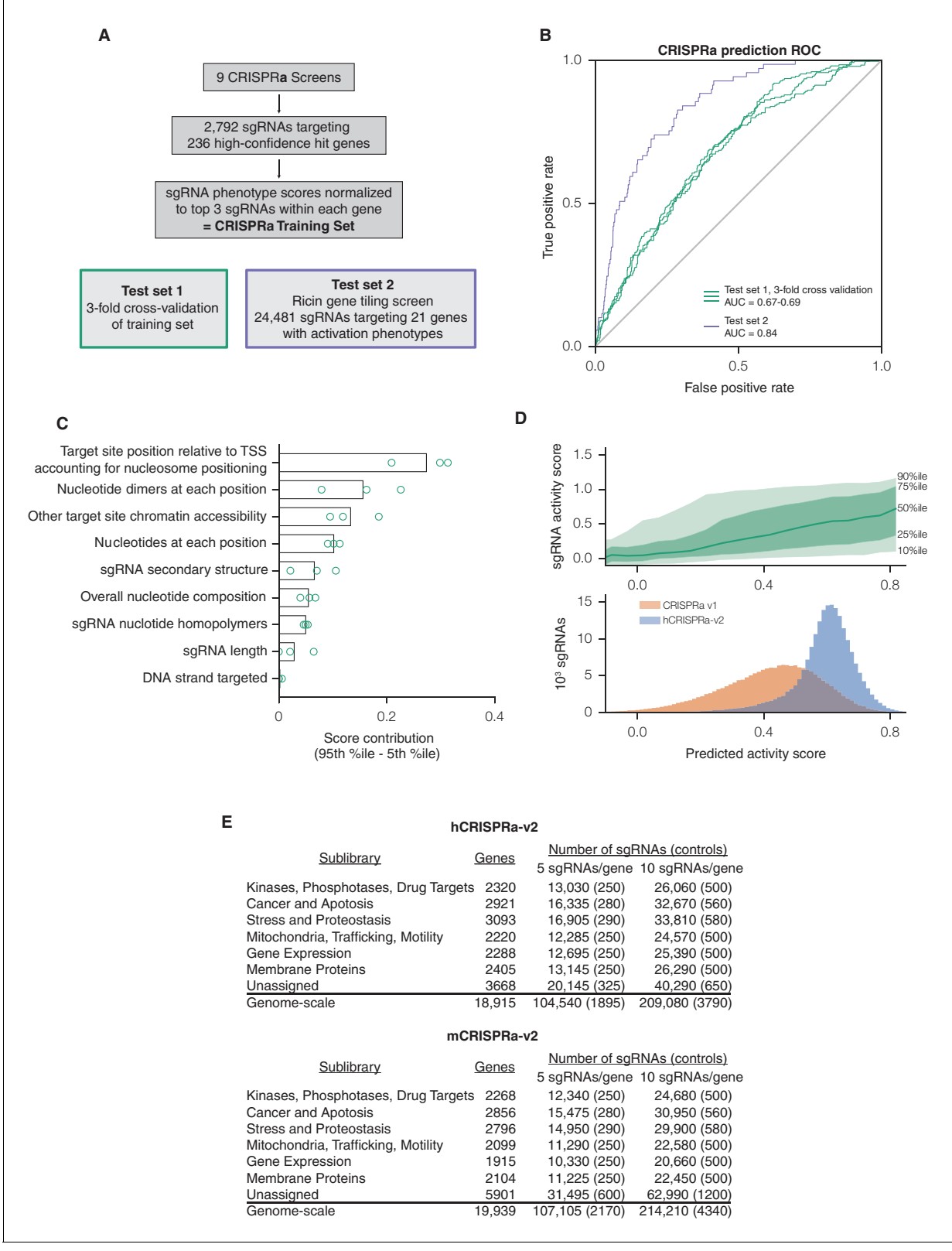

**Figure 2.** A machine learning approach for identifying highly active sgRNAs for CRISPRa. (**A**) Schematic of CRISPRa datasets. CRISPRa activity scores were generated from screen data and subjected to 3-fold cross-validation due to the smaller sample size. Ricin tiling data was limited to 21 genes that were previously shown to modulate sensitivity to ricin upon CRISPRa overexpression (*Gilbert et al., 2014*). (**B**) ROC analysis of machine learning approach using CRISPRa datasets, conducted as in *Figure 1B*. (**C**) Relative contribution of feature categories for CRISPRa, calculated as in *Figure 1C*. *Figure 2 continued on next page*

*Figure 2 continued*

(D) Distribution of predicted activity scores in next-generation CRISPRa libraries. (Top) Predicted CRISPRa activity correlates with empirical activity scores. For the 67%/33% division used to predict sgRNAs for the hCRISPRa-v2 library, predicted scores for the 33% test set were plotted against the empirical activity score. Activity score percentiles are from all sgRNAs within 0.25 of the indicated activity score. Predicted activity was highly correlated with activity, with a Pearson R of 0.41 ($p<10^{-38}$). (Bottom) Distribution of predicted activity scores for CRISPRa v1 and hCRISPRa-v2 as calculated by the hCRISPRa-v2 regression model. (E) Composition of hCRISPRa-v2 and mCRISPRa-v2 sublibraries.

The following figure supplements are available for figure 2:

**Figure supplement 1.** Relationship between CRISPRa activity and sgRNA position relative to the TSS as predicted by SVR.

**Figure supplement 2.** Individual sgRNA feature contributions to predicted CRISPRa activity.

---

sgRNA constant region would be expected to further increase this fraction (*Chen et al., 2013*; *Dang et al., 2015*).

In order to validate our hCRISPRi-v2 library design and directly compare its performance to our published v1 library screens (*Gilbert et al., 2014*), we conducted a screen for genes essential for robust growth in the chronic myeloid leukemia cell line K562. We calculated the growth phenotype (γ) for each sgRNA (*Bassik et al., 2013*; *Kampmann et al., 2013*) and averaged these values across two screen replicates (*Figure 3—figure supplement 1* and *Supplementary file 7*). We found that the hCRISPRi-v2 sgRNA growth phenotypes targeting the Evers et al. essential gene sets correlated with predicted activity as above (*Figure 3A*; Pearson R = −0.42, $p<10^{-21}$), and therefore designing the libraries based on the top predicted scores selected for highly active sgRNAs. To quantify the fraction of active sgRNAs in our genome-scale libraries, we performed sgRNA-level ROC analysis, ranking sgRNAs by growth phenotype γ and classifying them as true or false positives if they targeted essential or non-essential genes, respectively (*Evers et al., 2016*). This analysis showed that hCRISPRi-v2 was greatly enriched for active sgRNAs (*Figure 3B*), and in particular the top 5 sgRNA/ gene library contained 80% active sgRNAs at 5% false positives, comparable to the pilot nuclease library tested by Evers et al. This improvement was due to the significant reduction in the number of inactive sgRNAs rather than any difference in the noise as assessed by the background distribution of non-essential gene-targeting sgRNAs (*Figure 3—figure supplement 1B*). In some instances, as with the known essential gene *VCP*, the difference in sgRNA phenotypes between v1 and v2 libraries was likely attributable to the transition from Ensembl to FANTOM as the TSS annotation source (*Figure 3—figure supplement 1C*) (*Cunningham et al., 2015*; *FANTOM Consortium and the RIKEN PMI and CLST (DGT) et al., 2014*; *Harrow et al., 2012*). Taken together, the above observations indicate the lower fraction of effective sgRNAs in previous libraries was a result of the algorithm and the TSS annotation used rather than any intrinsic limitation of CRISPRi.

## The hCRISPRi-v2 library robustly identifies essential genes in precision-recall analysis

We next sought to evaluate whether the enrichment for active sgRNAs in hCRISPRi-v2 over CRISPRi v1 resulted in improved accuracy and confidence for calling hit genes. We analyzed the screens with a consistent pipeline, scoring genes both by assigning a phenotype based on the mean of the top 3 sgRNAs targeting the gene (by absolute value) and by calculating the Mann-Whitney p-value of all 10 sgRNAs compared to the negative control sgRNAs. We visualized these gene scores as a volcano plot, with the phenotype effect size on the x-axis and p-value on the y-axis (*Figure 3C* and *Supplementary file 8*). Many hit genes exhibited much stronger p-values, including a substantial fraction that reached the optimal p-value obtainable in the Mann-Whitney test, indicating that the phenotypes of all targeting sgRNAs for these genes were more pronounced than any of the 3,790 non-targeting control sgRNAs. We also modeled noise and off-target effects in the system by generating a large set of 'negative control genes' composed of randomly selected sets of 10 non-targeting sgRNAs and scoring these genes as we did for true genes. In order to classify genes as hits in the screen, we used a score that integrated effect size and statistical confidence and applied the same threshold to both v1 and v2 screens. While in both screens fewer than 0.21% of these negative control genes scored as hits by these criteria, representing a ~2% empirically estimated false

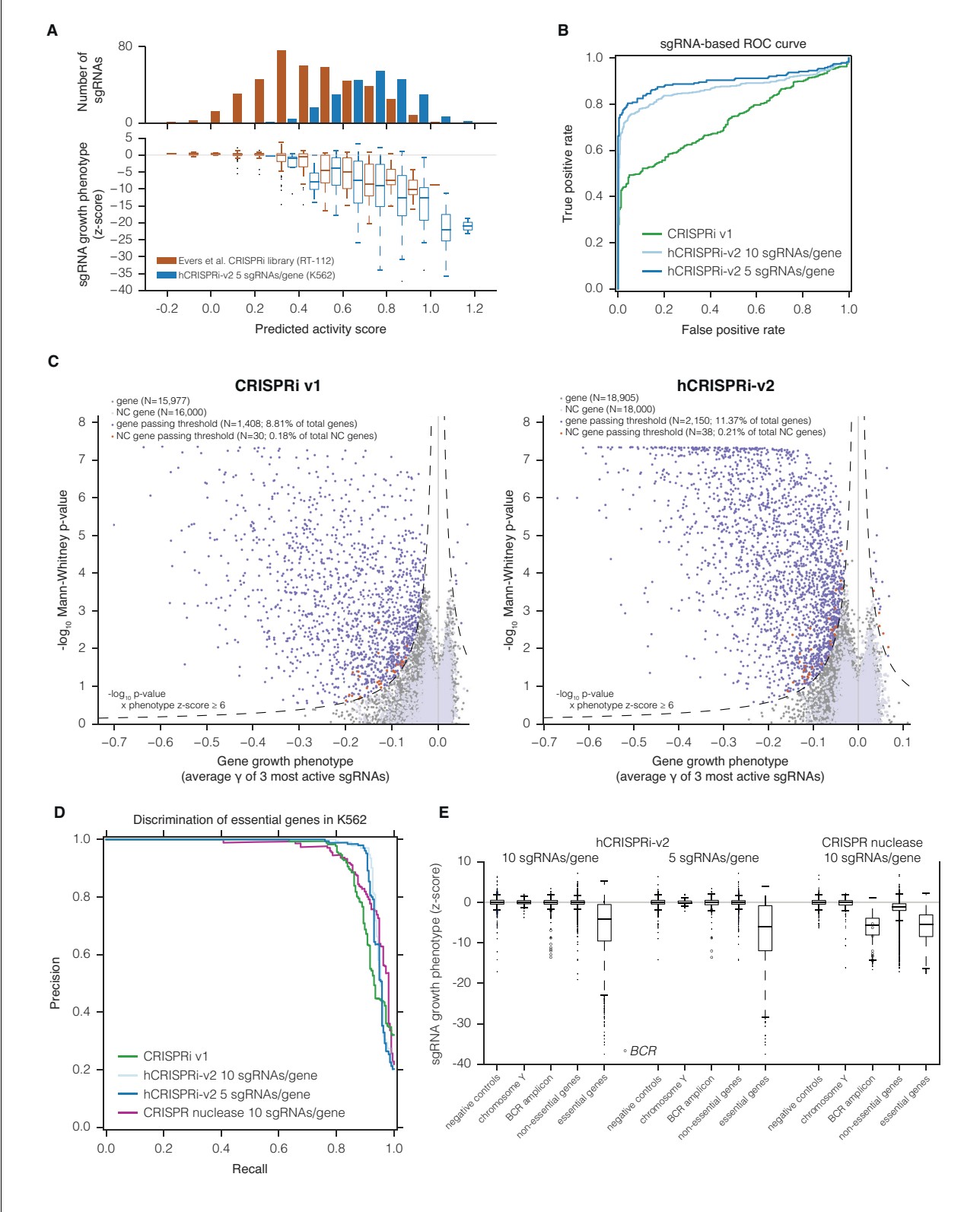

**Figure 3.** hCRISPRi-v2 outperforms CRISPRi v1 in screens for essential genes in K562. (**A**) Distribution and predicted scores for sgRNAs targeting essential genes. (Top) Predicted activity scores for sgRNAs from *Evers et al., 2016* or hCRISPRi-v2 targeting essential genes as defined by Evers et al. (*Evers et al., 2016*), binned in increments of 0.1. (Bottom) sgRNA growth phenotypes of the sgRNAs in the above bins, z-standardized to the distribution of sgRNAs targeting Evers et al. non-essential genes. (**B**) ROC analysis of sgRNAs from CRISPRi v1 or hCRISPRi-v2 targeting essential and

*Figure 3 continued on next page*

*Figure 3 continued*

non-essential genes. sgRNAs were ranked by γ, and considered true or false positives if they targeted essential or non-essential genes, respectively, as defined by Evers et al. (C) Volcano plots of gene phenotypes and p-values for growth screens performed with CRISPRi v1 (*Gilbert et al., 2014*) and hCRISPRi-v2. For each screen, genes phenotypes were calculated by averaging the growth phenotype (γ) of the 3 sgRNAs with the strongest γ by absolute value, and gene p-values were calculated by performing the Mann-Whitney test comparing all sgRNAs targeting the gene to the full set of negative control sgRNAs. For genes with multiple TSSs targeted, sgRNAs were grouped by TSS and the TSS with the lowest p-value was used for downstream analysis. A comparable number of negative control (NC) genes were generated by randomly sampling 10 non-targeting sgRNAs (with replacement) and analyzed as true genes. Empirically derived thresholds (dashed lines) were calculated as shown, using the NC gene distribution to derive the background standard deviation for z-score. (D) Precision-recall analysis of essential gene screens performed in K562. Statistical precision and recall of essential and non-essential gene sets (*Hart et al., 2014*) were calculated for genes ranked by growth phenotype in K562. For both CRISPRi and CRISPR nuclease screens (*Wang et al., 2015*), gene-level phenotypes were calculated as the average $\log_2$ fold-change of all sgRNAs targeting the gene (termed CRISPR Scores in ref. [*Wang et al., 2015*]). (E) Boxplots of CRISPRi and CRISPR nuclease sgRNA phenotypes for several gene sets. sgRNA γ scores (CRISPRi) or $\log_2$ enrichments (nuclease) were z-standardized to the corresponding negative control set. Boxplots display the distribution of the negative control sgRNAs or sgRNAs targeting genes on Y chromosome (excluding pseudo-autosomal genes), within the *BCR* amplicon, or in the gold standard essential sets used in (D). Individual phenotypes for sgRNAs targeting *BCR* are overlaid with the corresponding boxplot.

The following figure supplements are available for figure 3:

**Figure supplement 1.** sgRNA phenotypes from CRISPRi v1 and hCRISPRi-v2 growth screens.

**Figure supplement 2.** Precision-recall analysis of second-generation CRISPR nuclease essential gene screens.

discovery rate overall, we could confidently identify 2,150 essential gene hits in the v2 screen while our v1 screen identified 1,408 essential genes (*Figure 3C*).

In order to assess whether the stronger sgRNA- and gene-level growth phenotype γ scores produced by the hCRISPRi-v2 library resulted in improved discrimination of essential genes, we turned to precision-recall analysis of large 'gold standard' essential and non-essential gene sets introduced by Hart and colleagues (*Hart et al., 2014*). We ranked genes by their growth phenotype (γ) score and calculated at each phenotype threshold the trade-off between the recall of true essential genes and the avoidance of false positive non-essential genes, termed precision. In this analysis, the hCRISPRi-v2 library recalled over 91.2% of the gold standard essential genes at 95% precision compared to 81.5% in CRISPRi v1 (*Figure 3D*). We also found that the precision-recall of the top 5 sgRNA/gene half library was essentially identical to the full 10 sgRNA per gene library. Therefore, the 5 sgRNA/gene CRISPRi library represents a compelling tool for applications such as cell sorting-based screens (*Liberali et al., 2015*), or for *in vivo* screens where cell engraftment and library representation may represent a limiting factor (*Braun et al., 2016*; *Chen et al., 2015*).

Finally, we wanted to benchmark our hCRISPRi-v2 library against other recent genome-scale growth screens performed with nuclease-active Cas9 and novel second-generation libraries (*Doench et al., 2016*; *Hart et al., 2015*; *Wang et al., 2015*). Although these screens were performed in different labs and generally in different cell lines, precision-recall analysis offers a useful metric to compare these screens in an unbiased fashion (*Hart et al., 2014*). We found that our CRISPRi v2 library showed comparable or in many cases much greater discrimination (*Figure 3—figure supplement 2*). One published CRISPR nuclease screen was conducted in K562 (*Wang et al., 2015*) with a ~10 sgRNA/gene library, allowing for more direct comparison albeit still tempered by differences between labs and screening protocols. This screen recalled somewhat fewer (78.7% vs 90.8%) essential genes at 95% precision than our v2 library screen (*Figure 3D*). Together, these results indicate that our hCRISPRi-v2 library has a low false negative rate with few false positives, and the enrichment for highly active sgRNAs enables robust detection of phenotypes even with a compact library.

## CRISPRi does not induce non-specific toxicity at amplified genomic loci

We were also intrigued by the observation by Wang and colleagues of K562-specific essentiality of many genes neighboring the BCR-ABL translocation, which they demonstrated to be mediated by non-specific toxicity of CRISPR-induced double-stranded breaks in the amplified locus (*Wang et al., 2015*; *Wu et al., 1995*). This toxicity appears to be pervasive as similar effects have been observed

across a range of cancer cell lines (*Aguirre et al., 2016*; *Munoz et al., 2016*). To test whether this toxicity could be caused by CRISPRi as well, we used our hCRISPRi-v2 screen as a representative dataset. When we compared the phenotypes of CRISPRi sgRNAs to CRISPR nuclease at the *BCR* amplicon, with all phenotypes standardized to the distribution of negative controls to facilitate comparison, we found that sgRNAs targeting *BCR* were strongly depleted in both screens, as expected based on the critical role of the BCR-ABL fusion in this cancer cell line (*Naumann et al., 2001*), but few other CRISPRi sgRNAs in the region elicited growth defects (*Figure 3E*). We also found that CRISPR nuclease sgRNAs targeting the non-essential gene set were generally depleted relative to negative controls or chromosome Y-targeting sgRNAs, which should have no targets in the female-derived K562 cell line (*Klein et al., 1976*), suggesting that in a CRISPR screen Cas9 nuclease activity can lead to measurable toxicity not related to the function of individual genes but instead due to the formation of on-target DNA double strand breaks, even with alleles present only at 2–3 copies (*Naumann et al., 2001*). CRISPRi did not exhibit this generic toxicity at non-essential genes, allowing for detection of genes with subtle phenotypes relative to negative controls. Importantly, however, the vast majority of sgRNAs targeting essential genes showed a clear separation from the non-essential gene distribution (*Figure 3E*), demonstrating the high degree of sensitivity for detecting loss-of-function phenotypes with both CRISPR nuclease and CRISPRi screens.

## The hCRISPRa-v2 library identifies more genes that modify robust growth rates upon overexpression

Finally, we sought to validate our hCRISPRa-v2 library design by conducting a screen for growth phenotypes in K562 cells expressing SunTag-VP64 constructs (*Gilbert et al., 2014*; *Tanenbaum et al., 2014*). We conducted two replicate growth screens (*Figure 4—figure supplement 1A* and *Supplementary file 9*) and analyzed the screen as with the hCRISPRi-v2 screen above to directly compare the results to our published CRISPRa screens (*Gilbert et al., 2014*). Our hCRISPRa-v2 screen identified 540 genes to modify robust growth rates upon overexpression, 257 more than our previous CRISPRa screen (*Figure 4A* and *Supplementary file 10*). Beyond these additional hits, the v1 and v2 screens showed good agreement (*Figure 4—figure supplement 1B*), and the top categories in DAVID analysis of the v1 screen (*Huang et al., 2009*), enrichment for transcription factor genes (in particular homeobox and forkhead box transcription factors), received ~3-fold greater enrichment scores in the hCRISPRa-v2 hits (*Figure 4—figure supplement 1C*), indicating the strong biological coherence of the additional genes. Analysis of the sgRNA growth phenotype (γ) scores for genes that were hits in both v1 and v2 screens showed that a greater fraction of sgRNAs were highly active (69.3% in hCRISPRa-v2 with 5 sgRNAs/gene versus 45.1% in CRISPRa v1; *Figure 4C* and *Figure 4—figure supplement 1D*), further validating improvements in the library design. In addition, as with our CRISPRi results, several genes identified in the hCRISPRi-v2 screen but not in v1, including hematopoietically-expressed homeobox *HHEX* and forkhead box C1 *FOXC1*, could be attributed to the use of the CAGE-based FANTOM5 TSS annotation (*Figure 4D* and *Figure 4—figure supplement 1E*). Finally, we compared the growth phenotypes from hCRISPRi-v2 to hCRISPRa-v2 and found that the two methods identified non-overlapping sets of genes that modify robust growth (*Figure 4—figure supplement 1F*), consistent with our previous results and highlighting the complementary information provided by these two approaches.

## Discussion

Establishing design rules for effective reagents is critical to the implementation of genome-scale screening technologies. Our previous work established genome-scale CRISPRi and CRISPRa libraries as robust, specific, and complementary tools for dissecting biological pathways in human cells (*Gilbert et al., 2014*). Here, we significantly improve upon this technology by developing a comprehensive predictive model to accurately identify highly active sgRNAs. This model includes both features specific for CRISPRi and CRISPRa, such as positioning relative to the TSS, as well as features like nucleosome occupancy that we expect to be generally important for most Cas9-mediated applications (*Horlbeck et al., 2016*), and was able to accurately predict sgRNA activity in screen performed in a cell type it had not previously evaluated (*Evers et al., 2016*). We used this prediction algorithm to design four new genome-scale libraries targeting human and mouse genomes. These

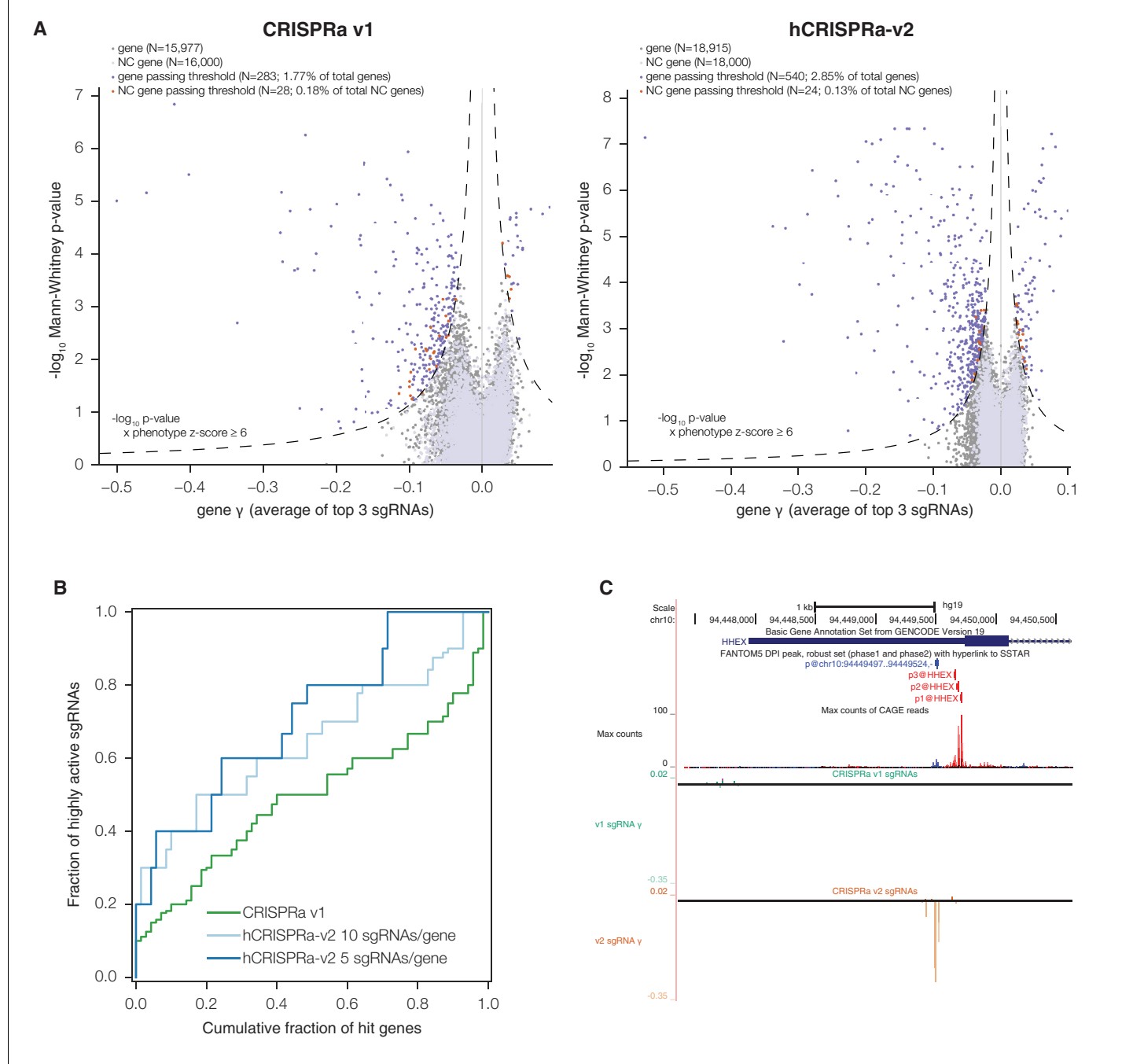

**Figure 4.** hCRISPRa-v2 outperforms CRISPRa v1 in screens for genes that modify growth rates upon overexpression. (**A**) Volcano plots of gene phenotypes and p-values for growth screens performed with CRISPRa v1 (*Gilbert et al., 2014*) and hCRISPRa-v2, presented as in *Figure 3C*. (**B**) Cumulative distributions of fraction of highly active sgRNAs targeting strong hit genes shared between CRISPRa v1 and hCRISPRa-v2 screens. Highly active sgRNAs for CRISPRa were defined as those with negative γ scores more than two standard deviations from the mean of non-targeting control sgRNAs (see *Figure 4—figure supplement 1D*). (**C**) UCSC Genome Browser tracks depicting TSS annotations and CRISPRa growth phenotypes for example gene *HHEX*.

The following figure supplement is available for figure 4:

**Figure supplement 1.** sgRNA phenotypes and gene category enrichment scores from CRISPRa v1 and hCRISPRa-v2 growth screens.

libraries are available on Addgene and *in silico* to facilitate design of focused libraries or targeted experiments (*Supplementary file 3–6*).

By performing CRISPRi and CRISPRa screens for genes that modify robust growth, we validate our sgRNA predictions and find that our hCRISPRi-v2 and hCRISPRa-v2 libraries represent a significant advance over our previous work in the fraction of highly active sgRNAs, the number of hits detected, and the statistical confidence of these hits. For CRISPRi, these improvements result in the accurate discrimination of essential genes by precision-recall analysis even with a compact 5 sgRNA/gene library. We believe that the greatly improved sgRNA prediction and lack of non-specific toxicity due to nuclease activity, combined with our previous findings that CRISPRi enables inducible, reversible, and homogenous manipulation of gene expression (*Gilbert et al., 2014*; *Mandegar et al., 2016*; *Qi et al., 2013*), make CRISPRi a state-of-the-art approach for loss-of-function studies.

While the libraries described here target genes annotated as protein-coding, CRISPRi and CRISPRa have been shown to be effective for repressing and activating transcription of non-coding genes as well (*Gilbert et al., 2014*; *Luo et al., 2016*; *Zhao et al., 2014*). Our v2 sgRNA prediction algorithms may enable the design of libraries to systematically manipulate expression of these transcripts as well. Furthermore, combining CRISPRi and CRISPRa with methods for robustly expressing multiple sgRNAs (*Kabadi et al., 2014*; *Wong et al., 2016*; *Zalatan et al., 2015*) will allow for simultaneous control of several genes, facilitating dissection of cellular pathways and systematic mapping of mammalian genetic interactions (*Bassik et al., 2013*; *Costanzo et al., 2010*; *Schuldiner et al., 2005*). Broadly, increased quantitative understanding of the factors dictating (d)Cas9 activity and specificity will greatly enhance the expanding set CRISPR-mediated technologies for controlling gene expression (*Hilton et al., 2015*; *Perez-Pinera et al., 2013*; *Vojta et al., 2016*), imaging targeted loci (*Chen et al., 2013*; *Shao et al., 2016*), or precisely editing the genome (*Komor et al., 2016*; *Tsai et al., 2014*).

## Materials and methods

### Machine learning for CRISPRi and CRISPRa sgRNA activity

Training and test datasets

The CRISPRi activity score dataset was obtained from *Horlbeck et al. (2016)*. CRISPRi and CRISPRa ricin tiling data was obtained from *Gilbert et al. (2014)*. CRISPRa activity scores were generated as previously described for the CRISPRi activity dataset, using data from 9 published and unpublished screening datasets. Hit genes were selected using the formula |effect size Z-score x $\log_{10}$ p-value| $\geq$ 20 in any screen, and the phenotypes for sgRNAs targeting each gene were extracted from the screen in which the gene was a hit and normalized to the mean of the top 3 sgRNAs by absolute value. All datasets are included here as *Supplementary file 1*.

Generating TSS annotations

In order to leverage the high accuracy of the FANTOM TSS annotations but remain compatible with the comprehensive, systematic, and established Ensembl transcript models, a hybrid approach was taken. First, the full set of protein-coding genes and transcripts were selected as previously described (*Gilbert et al., 2014*), using Ensembl release 74 (corresponding to genome assemblies hg19 for human and mm10 for mouse; RRID:SCR_002344) and the APPRIS pipeline to identify the relevant functional transcripts and establish a preliminary set of TSS annotations (*Cunningham et al., 2015*). FANTOM CAGE peak BED files (RRID:SCR_002678; Human: http://fantom.gsc.riken.jp/5/datafiles/phase1.3/extra/TSS_classifier/TSS_human.bed.gz, accessed March 2, 2015; Mouse: http://fantom.gsc.riken.jp/5/datafiles/phase1.3/extra/TSS_classifier/TSS_mouse.bed.gz, accessed June 28, 2015 and lifted over to mm10 coordinates with UCSC liftOver tool) were then used to refine this annotation. Specifically, for each gene, the TSS was identified as the same-stranded peaks within 30 kb of any Ensembl TSS that matched the gene symbol and was labeled 'p1@gene' (here referred to as 'primary TSS') or 'p2@gene' ('secondary TSS'; included only where the peak passed the FANTOM robust threshold for TSS), and the annotation support was labeled as 'CAGE, matched peaks.' If multiple matches were found the closest TSS to a known Ensembl TSS was used. If no match was found, the gene could then be matched with any same-stranded CAGE peak within 500 bp labeled

as 'p1' or 'p2,' and annotation support was considered 'CAGE, primary peaks.' In the above cases, primary and secondary TSSs were targeted separately (i.e. by 10 sgRNAs each) if they were farther than 1 kb apart, or together as one 'P1P2' TSS. Where no matched or un-matched primary peaks were found, TSS annotations could be refined by any robust peaks or permissive peaks within 200 bp of the annotation (labeled 'CAGE, robust peak' or 'CAGE, permissive peak') or simply use the Ensembl/APPRIS annotation where no CAGE support was available ('Annotation'). This combined annotation is included as *Supplementary file 2*.

## Calculating features

Position relative to the primary and secondary TSSs was calculated from the genomic coordinate of the 3'G of the PAM for each sgRNA to the upstream and downstream edge of each TSS range. Sequence features of the sgRNA and target sites were determined using custom scripts written in Python 2.7 (RRID:SCR_008394) with the Biopython module (RRID:SCR_007173) (*Cock et al., 2009*). The hCRISPRi-v2.1 algorithm included a change in these scripts to fix how certain sequence homopolymers were scored. All other libraries were generated with this improvement incorporated, and all analysis in this paper was performed with the v2.1 algorithm. RNA folding metrics were calculated using the ViennaRNA package (RRID:SCR_008550; version 2.2.5) with default parameters (*Lorenz et al., 2011*). Chromatin features at the target site were calculated as described previously (*Horlbeck et al., 2016*), averaging the signal at each base of the target site including the PAM. Custom Python scripts with the module bxpython (v0.5.0, https://github.com/bxlab/bx-python) to extract the processed continuous signal from the following BigWig files obtained from the ENCODE consortium: MNase-seq https://www.encodeproject.org/files/ENCFF000VNN/ (Michael Snyder lab, Stanford University), DNase-seq https://www.encodeproject.org/files/ENCFF000SVY/ (Gregory Crawford lab, Duke University), and FAIRE-seq https://www.encodeproject.org/files/ENCFF000TLU/ (Jason Lieb lab, University of North Carolina) (*ENCODE Project Consortium, 2012*). Beyond the nucleosome positioning information incorporated in the sgRNA positioning learning models, no chromatin data was used for predicting sgRNA activity for the mouse genome.

## Machine learning

Training and test activity score sets were first divided into 80%/20% or 67%/33% sets of genes for CRISPRi or CRISPRa, respectively, as described in the results section. The training set parameters were then transformed according to their distribution as depicted in *Figure 1A*. For binning parameters, a fixed width was chosen for each feature and applied over the range of values, with the uppermost and lower-most bins collapsed with the neighboring bins if the number of data points at each edge were sparse. Each feature was then split into individual parameters for each bin and sgRNAs were assigned a 1 for the bin if the value fell within the bin or 0 if not. For linearizing sgRNA positioning parameters with continuous curves, sgRNA positions were fit to the activity score (individually for the distance to each TSS coordinate) using SVR with a radial basis function kernel and hyperparameters C and gamma determined using a grid search approach cross-validated within the training set. The fit score at each position was then used as the transformed linear parameter. Binary parameters were assigned a 1 if true or a 0 if false. All linearized parameters were z-standardized and fit with elastic net linear regression, with the l1/l2 ratio set by cross-validated grid search. All machine learning and downstream analysis was performed with custom Python scripts and the scikit-learn suite, version 0.15.0 (RRID:SCR_002577) (*Pedregosa et al., 2011*).

## Design of next-generation CRISPRi and CRISPRa libraries

### Prediction of sgRNA activity scores

All sequences within −25 and 500 bp (for CRISPRi) or −550 and −25 bp (for CRISPRa) of the upstream or the downstream edge of the primary or secondary TSS and containing 19 bp followed by an NGG PAM were extracted as potential sgRNAs for downstream scoring of predicted activity. All sequences were prepended with a 5' G to enable robust transcription from the U6 promoter, whether or not this base was present in the genomic sequence. Parameters were calculated for all sgRNAs as above, and transformed and scored using the CRISPRi or CRISPRa regression model from an arbitrarily chosen training set (test set ROC-AUC corresponding to these sets reported in results section).

## Off-target scoring

Prediction of sgRNA off-target effects was performed using weighted Bowtie (v1.0.0, RRID:SCR_005476 [*Ben Langmead et al., 2009*]) alignment largely as previously described (*Gilbert et al., 2014*) with several adjustments. The '–tryhard' flag was added to the Bowtie command to increase sensitivity for mismatched sgRNA target sites. The hg19 and mm10 genomes used for alignment were masked at mitochondrial sequences and pseudoautosomal sequence to eliminate 'false positive' multiple alignments. Most importantly, as CRISPRi and CRISPRa have maximal effects proximal to the TSS, potential off-target alignments in these regions now were prioritized by creating a reference sequence corresponding to 1 kb windows around each TSS as defined above, along with the 5' end of every Ensembl transcript annotation. Reference sequences were generated using bedtools (v2.17.0, RRID:SCR_006646 [*Quinlan and Hall, 2010*]). In order to pass at the strictest threshold, sgRNAs were required to have no more than 1 alignment (the sgRNA target site itself) with 'mismatch score' (*Gilbert et al., 2014*) less than 31 proximal to the TSS and under 21 in the genome. (For hCRISPRi-v2, 96.6% of sgRNAs incorporated passed at this threshold.) In cases of difficult to target genes or close gene families, sgRNAs were allowed at relaxed thresholds. In descending order, these were: 1 alignment under 31 proximal to the TSS (no genomic threshold), 1 alignment under 21 in the genome, 2 alignments under 31 proximal to the TSS, and 3 alignments under 31 proximal to the TSS.

## sgRNA selection

sgRNAs were chosen for inclusion into the genome-scale libraries based on predicted activity scores, empirical activity scores where available, off-target filtering, overlap with other sgRNAs already selected, and sequences with no restriction sites for enzymes used in cloning or sequencing sample processing (BstXI, BlpI, and SbfI). Empirical activity scores for CRISPRi/a v1 sgRNAs were generated as for the training sets above at a lower discriminant threshold of 7, and the corresponding sgRNAs were standardized to 19 bp with a 5' G preprended as above and subjected to the same revised off-target scoring procedure. For each TSS targeted by the library, up to 2 sgRNAs with the strongest empirical evidence were included first if the empirical activity score was at least 0.75, the sgRNA was less that 5 kb from the revised v2 TSS, the sgRNA passed the most stringent off-target filter, the sgRNA plus flanking cloning sequences did not contain extra restriction sites, and the sgRNA target site was at least 3 bp shifted from any previously selected target sequence. Once 2 empirically validated v1 sgRNAs were included, further sgRNAs fitting these criteria were not included but their predicted activity scores were increased by 0.2 to reflect the balance of information from the algorithm and empirical activity. Predicted sgRNAs were then sorted by best predicted activity score and included if the sgRNA passed the most stringent off-target filter, the sgRNA plus flanking cloning sequences did not contain extra restriction sites, the sgRNA target site was at least 3 bp shifted from any previously selected target sequence, and no more than 10 sgRNAs had been selected for the TSS. If fewer than 10 sgRNAs were selected by this algorithm, off-target stringency was iteratively relaxed as above and selection was continued to attain 10 sgRNAs. If 10 sgRNAs passing the most relaxed threshold could not be identified, the TSS was not targeted by the library.

sgRNA on-target and off-target prediction algorithms, library design scripts, and associated files are available at https://github.com/mhorlbeck/CRISPRiaDesign.

## Negative controls

For each library, the frequency of each DNA base at each position along the sgRNA protospacer sequence was calculated. Random sgRNA protospacer sequences weighted by these base frequencies were then generated to mirror the composition of the targeting sgRNAs. These were then filtered for sgRNAs with 0 alignments with a mismatch score less than 31 proximal to the TSS and 0 alignments under 21 in the genome as above.

## Library cloning

Protospacer sequences were appended with cloning sequences and then unique PCR adapters corresponding to the designated sublibrary. The half libraries were determined from the first 5 and second 5 sgRNAs selected into the library for each TSS according to the algorithm above. Genes were then partitioned into one of the 13 sublibraries defined by (*Kampmann et al., 2015*), compressed

into the indicated 7 groupings. Each sgRNA was ordered as two oligonucleotide sequences to produce a narrower distribution of sgRNA representation. Overall, oligo sequences were 84 bp and had the following format:

```
 5'  pcr adapter        BstXI       protospacer         BlpI    3'  pcr adapter
==================CCACCTTGTTGGNNNNNNNNNNNNNNNNNNNNNGTTTAAGAGCTAAGCTG==================
                            ?|||||||||||||||||||||
Genomic sequence: .............NNNNNNNNNNNNNNNNNNNNNNNGG...............
```

Oligonucleotides were synthesized by Agilent Technologies (RRID:SCR_013575; Santa Clara, CA) on 244K oligo arrays, and cloned into the sgRNA expression vector as previously described (*Gilbert et al., 2014*). The library sgRNA expression vector 'pCRISPRia-v2' was identical to the CRISPRi/a v1 plasmid (pU6-sgRNA EF1Alpha-puro-T2A-BFP, Addgene #60955) with the addition of two SbfI restriction sites used for sequencing sample processing.

## Genome-scale CRISPRi and CRISPRa screen for essential genes

The screens for genes required for robust growth were conducted essentially as previously described (*Gilbert et al., 2014*). Briefly, plasmid libraries were packaged into lentivirus in HEK293T cells (RRID:CVCL_0063) and infected into a previously established polyclonal K562 cell line stably expressing dCas9-KRAB grown in 3L spinner flasks (Bellco, Vineland, NJ). After two days, infected cells were selected with 0.75 µg/mL puromycin (Tocris, Bristol, UK) for two days, allowed to recover for one day, and then cultured at a minimum of $750 \times 10^6$ cells in 1.5L standard media (RPMI-1640 with 10% Fetal Bovine Serum and 1x supplemental glutamine, penicillin, and streptomycin) from 'T0' to 'endpoint,' determined by ~10 cell doublings after T0. CRISPRi screen cells were mock-treated with 0.1% DMSO (Sigma-Aldrich, St. Louis, MO) but otherwise left untreated. Screens were performed as independent replicates starting from the infection step. The K562 dCas9-KRAB and SunTag-VP64 cell lines were obtained from (*Gilbert et al., 2014*) and had been constructed from K562 cells originally obtained from ATCC (RRID:CVCL_0004). Cytogenetic profiling by array comparative genomic hybridization (not shown) closely matched previous characterizations of the K562 cell line (*Naumann et al., 2001*). All cell lines tested negative for mycoplasma contamination (MycoAlert Kit, Lonza, Basel, Switzerland) in regular screenings.

Frozen samples of $250 \times 10^6$ cells collected at T0 and endpoint were processed as previously described (*Gilbert et al., 2014*), with the substitution of an SbfI restriction digest (SbfI-HF, New England Biolabs, Ipswich, MA) in place of the MfeI digest in the genomic DNA fragmentation and enrichment step. The sgRNA-encoding regions were sequenced on an Illumina HiSeq-4000 using custom primers. Sequencing reads were aligned to the expected library sequences using Bowtie (v1.0.0, [*Ben Langmead et al., 2009*]) and read counts were processed using custom Python scripts (available at https://github.com/mhorlbeck/ScreenProcessing) based on previously established shRNA screen analysis pipelines (*Bassik et al., 2013*; *Kampmann et al., 2013*). sgRNAs represented with fewer than 50 sequencing reads in both T0 and Endpoint were excluded from analysis. sgRNA growth phenotypes (γ) were calculated by normalizing sgRNA $\log_2$ enrichment from T0 to endpoint samples and normalizing by the number of cell doublings in this time period. CRISPRi v1 screen data from *Gilbert et al. (2014)* was re-analyzed using this pipeline, and the hCRISPRi/a-v2 5 sgRNA/gene libraries were evaluated by analyzing the sgRNA read counts corresponding to only the 5 sgRNA/gene sublibraries. Gene ontology analysis was conducted using DAVID 6.7 (*Huang et al., 2009*) with default search categories and with background lists representing the genes targeted by the CRISPRa v1 or hCRISPRa-v2 libraries where appropriate. For *Figure 4B* and *Figure 4—figure supplement 1D*, 'shared hit' genes were 70 genes that scored as strong anti-growth hits (phenotype z-score x − $\log_{10}$ p-value $\leq -10$) in both CRISPRa v1 and hCRISPRa-v2.

## Acknowledgements

We would like to thank Dr. Manuel Leonetti, Ben Barsi-Rhyne, Dr. Jonathan Friedman, and Dr. Jodi Nunnari for generously sharing unpublished screening data for determination of sgRNA activity. We would also like to thank Dr. Xuebing Wu and members of the Weissman lab, particularly Dr. Joshua Dunn and Manny DeVera, for helpful discussions and assistance. We thank Dr. Laurakay Bruhn, Dr.

Daniel Ryan, Dr. Luke Fairbairn, and Dr. Peter Tsang of Agilent Technologies for their assistance on the design and synthesis of oligonucleotide pools. MAH, LAG, JEV, BA, YC, APF, and JSW were supported by the Howard Hughes Medical Institutes and the National Institutes of Heath (P50 GM102706, U01 CA168370, R01 DA036858). LAG was supported by the Leukemia and Lymphoma Society. RAP, CYP, and JEC were supported by the Li Ka Shing Foundation. MK was supported by NIH/NIGMS DP2 GM119139.

## Additional information

### Competing interests

MAH: patent application related to CRISPRi and CRISPRa screening (PCT/US15/40449). JSW is a founder of, and MAH and LAG are consultants for, KSQ Therapeutics, a CRISPR functional genomics company. LAG, JSW: filed a patent application related to CRISPRi and CRISPRa screening (PCT/US15/40449). JSW is a founder of, and MAH and LAG are consultants for, KSQ Therapeutics, a CRISPR functional genomics company. MK: patent application related to CRISPRi and CRISPRa screening (PCT/US15/40449). The other authors declare that no competing interests exist.

### Funding

| Funder | Grant reference number | Author |
|---|---|---|
| Howard Hughes Medical Institute | | Max A Horlbeck<br>Luke A Gilbert<br>Jacqueline E Villalta<br>Britt Adamson<br>Yuwen Chen<br>Alexander P Fields |
| National Institutes of Health | P50 GM102706 | Max A Horlbeck<br>Luke A Gilbert<br>Jacqueline E Villalta<br>Britt Adamson<br>Yuwen Chen<br>Alexander P Fields |
| National Institutes of Health | U01 CA168370 | Max A Horlbeck<br>Luke A Gilbert<br>Jacqueline E Villalta<br>Britt Adamson<br>Yuwen Chen<br>Alexander P Fields |
| National Institutes of Health | R01 DA036858 | Max A Horlbeck<br>Luke A Gilbert<br>Jacqueline E Villalta<br>Britt Adamson<br>Yuwen Chen<br>Alexander P Fields |
| Leukemia and Lymphoma Society | | Luke A Gilbert |
| National Cancer Institute | Pathway to Independence Award, K99 CA204602 | Luke A Gilbert |
| Li Ka Shing Foundation | | Ryan A Pak<br>Chong Yon Park<br>Jacob E Corn |
| National Institute of General Medical Sciences | DP2 GM119139 | Martin Kampmann |

The funders had no role in study design, data collection and interpretation, or the decision to submit the work for publication.

### Author contributions

MAH, Conceived of and conducted algortihm development and data analysis, contributed to genome-scale screening experiments, and wrote this report; LAG, Contributed to sgRNA library development and genome-scale screening experiments, and helped write this report; JEV,

Contributed to sgRNA library development, generated sgRNA libraries, and contributed technical assistance; BA, Conducted genome-scale screening experiments; RAP, Generated sgRNA libraries; YC, Contributed technical assistance; APF, Contributed to algorithm development; CYP, Supervised sgRNA library generation and conducted genome-scale screening experiments; JEC, Contributed to algorithm development and supervised sgRNA library generation; MK, Contributed to and supervised algorithm development; JSW, Conceived of and supervised the study, and helped write this report

### Author ORCIDs
Max A Horlbeck, http://orcid.org/0000-0002-3875-871X
Ryan A Pak, http://orcid.org/0000-0003-3507-3122
Jacob E Corn, http://orcid.org/0000-0002-7798-5309
Martin Kampmann, http://orcid.org/0000-0002-3819-7019
Jonathan S Weissman, http://orcid.org/0000-0003-2445-670X

## Additional files

### Supplementary files
• Supplementary file 1. CRISPRi and CRISPRa activity score datasets.

• Supplementary file 2. TSS annotations for hg19 and mm10 genomes.

• Supplementary file 3. Library composition of hCRISPRi-v2 and hCRISPRi-v2.1.

• Supplementary file 4. Library composition of mCRISPRi-v2.

• Supplementary file 5. Library composition of hCRISPRa-v2.

• Supplementary file 6. Library composition of mCRISPRa-v2.

• Supplementary file 7. sgRNA read counts and growth phenotypes for hCRISPRi-v2 screens performed in K562.

• Supplementary file 8. Gene growth phenotypes and p-values for hCRISPRi-v2 screens performed in K562.

• Supplementary file 9. sgRNA read counts and growth phenotypes for hCRISPRa-v2 screens performed in K562.

• Supplementary file 10. Gene growth phenotypes and p-values for hCRISPRa-v2 screens performed in K562.

### Major datasets
The following previously published datasets were used:

| Author(s) | Year | Dataset title | Dataset URL | Database, license, and accessibility information |
|---|---|---|---|---|
| FANTOM Consortium | 2014 | A promoter-level mammalian expression atlas | http://fantom.gsc.riken.jp/5/datafiles/phase1.3/extra/TSS_classifier/ | Publicly available at FANTOM (files TSS_human.bed.gz, TSS_mouse.bed.gz) |

| ENCODE Consortium/Snyder | 2011 | Determinants of nucleosome organization in primary human cells | https://www.encodeproject.org/files/ENCFF000VNN | Publicly available at ENCODE (accession no. ENCFF000VNN) |
| ENCODE Consortium | 2012 | An integrated encyclopedia of DNA elements in the human genome | https://www.encodeproject.org/files/ENCFF000TLU | Publicly available at ENCODE (accession no. ENCFF000TLU) |

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
