## [Decision Letter]

Thank you for submitting your article "Compact and highly active next-generation libraries for CRISPR-mediated gene repression and activation" for consideration by *eLife*. Your article has been favorably evaluated by Jessica Tyler (Senior Editor) and two reviewers, one of whom is a member of our Board of Reviewing Editors. The reviewers have opted to remain anonymous.

The reviewers have discussed the reviews with one another and the Reviewing Editor has drafted this decision to help you prepare a revised submission.

Summary:

Pooled CRISPR screenings have become important methodologies to quickly associate genes with their functions in a high-throughput fashion. These platforms include gene knockout screens, gene inactivation (CRISPRi) and activation (CRISPRa). For such powerful technologies, it is important to further boost their efficacy and minimize any potential off-target effect. This manuscript deals with this very issue, and describes a systematic analysis of features that enable effective short guide RNA (sgRNA) activity in CRISPRi or CRISPRa screens. The result is new, highly effective libraries containing 5 or 10 sgRNAs per gene. These new libraries are compact and will be powerful tools for discovery of gene function moving forward.

Specifically, the manuscript details the use of machine learning algorithms and other informatic approaches to determine what features render a sgRNA most effective. Key aspects involve targeting a nucleosome-depleted region and the position from the observed transcription start site (using the more accurate FANTOM positions for TSS designation). These properties, coupled with specific sequence features can be combined to predict very active sgRNAs. The authors do a great job demonstrating the effectiveness of their predictions and the new CRISPRi libraries. This clearly establishes the promise of CRISPRi as a potent methodology that avoids a number of the pitfalls that using catalytically active CRISPR/Cas9 entails.

To facilitate dissemination of this knowledge, they have made the libraries available on Addgene (in 5 or 10 sgRNA per gene varieties) that target mouse or human genes. Further, they share the sequences of the best sgRNAs in supplemental tables, additionally increasing the impact and breadth of this work.

In short, this is a nicely written story with convincing data. I have only a few changes to suggest to the text and display items. These are aimed at making the manuscript maximally helpful to people in the field who might be designing their own guides against non-coding RNAs, as well as increasing the interest for those who are just curious about how CRISPRi works.

Essential revisions:

1) Authors claimed that the next-generation libraries for CRISPR-mediated gene repression and activation have higher activity than the old version. The experimental demonstration for new version of CRISPRa is completely missing. It's therefore premature to make such statement in the title and in the context. This issue should be clarified.

2) The new version of library outperform the old version, mostly based on statistic prediction and analysis. However, one might argue that the original library is "good enough" with "sub-optimal" performance in the identification of genes' function in majority of cases. It would be much more convincing if they could show examples of genes identified from new libraries that would be missing in the old fashion way. In this case, comprehensive validation of these candidate genes are needed to make such a claim.

3) Authors said that the new CRISPRi screen has undetectable non-specific toxicity seen with CRISPR nuclease approaches. This is actually the feature of CRISPRi, not the new design of CRISPRi library. It's misleading to give such credit to the new algorithm of design.

4) Likewise, we couldn't quite get the point why the new library would have improved off-target effects. It's understandable that the new design might improve the on-target activity. As to the off-target rate, it's puzzling to understand the mechanism behind this observation.

5) Although the effects of nucleosomes (DNAse, MNase, FAIRE) on positioning of sgRNAs for CRISPRi is strong, the importance for CRISPRa seems quite modest from the data presented- with position relative to the TSS being much more important. Why might this be? I am not convinced that this means that activating guide RNAs are indifferent to nucleosomes (and the authors don't suggest this). So why the difference? It would be helpful to the community to get some comment on this. As the manuscript stands, one could interpret the findings to say that one needed to target nucleosome-depleted regions to inhibit gene expression, but that activating guides could more readily penetrate chromatin, which seems unlikely. Could the authors clarify?

One possibility is that the optimal location upstream of the TSS (which appears to be -100 to -200 upstream) is typically nucleosome depleted at most active genes and so there isn't much dynamic range in the nucleosome signal detected in this region. This would compress the information you could get out of this parameter, perhaps making it seem less important because there was less variability among genes.

6) Given the above distinction, it would be preferable to show the score contribution for 'target site position relative to TSS' and 'target site chromatin accessibility' separately for CRISPRi (Figure 1) as well as CRISPRa (Figure 2).

7) Can the authors comment on the very strong peak of effectiveness for CRISPRi sgRNAs just downstream of the TSS? This likely reflects something in addition to nucleosome-deprivation as being helpful for CRISPRi. I find the very sharp peak in maximal activity in Figure 1—figure supplement 1 to be really striking (even in comparison to previous work using less well-refined TSSs), and would encourage its inclusion in the final manuscript.

As the authors have noted before, CRISPRi really works best when the guide is positioned just downstream of the start site- perhaps because it is more effective to block early transcription elongation and the release from Pol2 pausing, rather than farther downstream once Pol2 is loaded with the machinery to plow through chromatin etc.?

For those working to block expression of novel or non-coding RNAs, we think it is worth getting this idea out there in a super-clear and obvious way- that the sweet spot for optimal guides is right downstream of the promoter.

8) Regarding the basic sgRNA prediction algorithms developed, will these be shared upon request? We see a github site for the tool created to assess chromatin features, but not for the broader prediction platform. Could the authors indicate in the Methods how an interested user might get help with sgRNA predictions?

---

## [Author Response]

*In short, this is a nicely written story with convincing data. I have only a few changes to suggest to the text and display items. These are aimed at making the manuscript maximally helpful to people in the field who might be designing their own guides against non-coding RNAs, as well as increasing the interest for those who are just curious about how CRISPRi works.*

*Essential revisions:*

*1) Authors claimed that the next-generation libraries for CRISPR-mediated gene repression and activation have higher activity than the old version. The experimental demonstration for new version of CRISPRa is completely missing. It's therefore premature to make such statement in the title and in the context. This issue should be clarified.*

We have now experimentally validated our hCRISPRa-v2 library by conducting a screen for genes that affect robust cell growth upon overexpression with CRISPRa. These data are presented in Figure 4 and the accompanying supplemental figure, and discussed in a new Results section. As with our hCRISPRi-v2 validation screen, we find that our next generation library outperforms CRISPRa v1 in number of hit genes identified (60% more hits and greater enrichment for functional categories; Figure 4), enrichment for highly active sgRNAs targeting each gene (Figure 4), and identification of new hit genes missed due to TSS mis-annotation (Figure 4 and Figure 4—figure supplement 1).

*2) The new version of library outperform the old version, mostly based on statistic prediction and analysis. However, one might argue that the original library is "good enough" with "sub-optimal" performance in the identification of genes' function in majority of cases. It would be much more convincing if they could show examples of genes identified from new libraries that would be missing in the old fashion way. In this case, comprehensive validation of these candidate genes are needed to make such a claim.*

We share the reviewers’ view that improvement of reagents should result in better identification of gene function for such an improvement to be of practical value. We feel that our data demonstrate the tangible benefits of the hCRISPRi-v2 library over the original CRISPRi v1 in several ways:

A) The hCRISPRi-v2 library recalls a further 19 of the 218 gold standard essential genes (Hart et al., 2014) at 95% precision than CRISPRi v1 (Figure 3). Of these 19 genes, 16 were also identified by the K562 CRISPR nuclease screen (Wang et al., Science 2015), providing further evidence for the validity of the gold standard gene set.

B) Similarly, the hCRISPRi-v2 library with just 5 sgRNAs per gene identifies 15 more of the gold standard essential genes than CRISPRi v1. That screen performance is improved over v1 with half the library size is a marked advance for many screening approaches where the scale of cell culture, FACS, etc. is limiting.

C) One essential gene included in the gold standard set, *VCP*, was not identified in our CRISPRi v1 screen but was a strong hit with our hCRISPRi-v2 growth screen. This improvement is likely due both to improved sgRNA selection and the use of the FANTOM TSS annotation, and is included as Figure 3—figure supplement 1.

*3) Authors said that the new CRISPRi screen has undetectable non-specific toxicity seen with CRISPR nuclease approaches. This is actually the feature of CRISPRi, not the new design of CRISPRi library. It's misleading to give such credit to the new algorithm of design.*

The non-specific toxicity mediated by CRISPR nuclease at amplified loci has only recently been reported by several groups (Wang et al., Science 2015; Aguirre et al., Cancer Discovery 2016; Munoz et al., Cancer Discovery 2016), but to our knowledge has not yet been investigated for CRISPRi. Therefore, we felt it timely to examine our CRISPRi data for this effect, and indeed found no toxicity at amplified or normal copy number loci in contrast to CRISPR nuclease. We agree with the reviewers that the undetectable toxicity due to DNA cleavage is very likely to be a general feature of CRISPRi, and by using the hCRISPRi-v2 results we merely intended to demonstrate this feature in a comprehensive way. We have now revised the text to clarify this point (Abstract; Results subsection “The hCRISPRa-v2 library identifies more genes that modify robust growth rates upon overexpression”).

*4) Likewise, we couldn't quite get the point why the new library would have improved off-target effects. It's understandable that the new design might improve the on-target activity. As to the off-target rate, it's puzzling to understand the mechanism behind this observation.*

We did make several adjustments to our off-target scoring algorithm that we expected to improve our ability to predict relevant off-target sites, most notably an increase in the sensitivity used in our Bowtie alignments and a more stringent filter for sgRNAs binding at off-target sites near TSSs (and thus more likely to cause off-target effects with CRISPRi and CRISPRa). However, the adjustments to our off-target predictions were a minor component of our design and indeed the off-target effects were quite low in the v1 as documented in Gilbert et al., 2014. We have removed references to this improvement in the Abstract and we have revised our language on off-target effects to reflect that this represents an algorithmic change rather than a major practical improvement (Results, Methods).

*5) Although the effects of nucleosomes (DNAse, MNase, FAIRE) on positioning of sgRNAs for CRISPRi is strong, the importance for CRISPRa seems quite modest from the data presented- with position relative to the TSS being much more important. Why might this be? I am not convinced that this means that activating guide RNAs are indifferent to nucleosomes (and the authors don't suggest this). So why the difference? It would be helpful to the community to get some comment on this. As the manuscript stands, one could interpret the findings to say that one needed to target nucleosome-depleted regions to inhibit gene expression, but that activating guides could more readily penetrate chromatin, which seems unlikely. Could the authors clarify?*

*One possibility is that the optimal location upstream of the TSS (which appears to be -100 to -200 upstream) is typically nucleosome depleted at most active genes and so there isn't much dynamic range in the nucleosome signal detected in this region. This would compress the information you could get out of this parameter, perhaps making it seem less important because there was less variability among genes.*

*6) Given the above distinction, it would be preferable to show the score contribution for 'target site position relative to TSS' and 'target site chromatin accessibility' separately for CRISPRi (Figure 1) as well as CRISPRa (Figure 2).*

In modeling the effects of chromatin accessibility, we incorporated sequencing-based readouts (DNase, MNase, FAIRE) as additional support but found that the strongest predictor of sgRNA activity was distance to the FANTOM-annotated TSS, which naturally convolutes both the stereotyped pattern of nucleosome positions (Jiang and Pugh, Nature Reviews Genetics 2009) as well as the effectiveness of CRISPRi/a effector domains relative to the TSS. Beyond the predictive value of this relationship, FANTOM TSS annotations are derived from CAGE-seq data from hundreds of cell lines and primary tissues, enabling us to reliably predict active sgRNAs across cell lines rather than relying on the nucleosome positioning in a particular cell line for which it happened to be measured. The applicability of our sgRNA predictions across cell lines is tested in Figure 3.

In using support vector regression to fit the patterns for CRISPRi and CRISPRa relative to the TSS and predict sgRNA activity (Figure 1—figure supplement 1 and Figure 2—figure supplement 1), however, the effects of nucleosome positioning and distance to TSS are convoluted such that the components difficult to evaluate separately. We have amended the axis labels on Figure 1 and Figure 2 to clarify the information obtained from TSS positioning and the other sequencing-based chromatin measurements.

We share with the reviewers’ observation that the periodic relationship between CRISPRa activity and distance to the TSS is less pronounced than for CRISPRi (Figure 2—figure supplement 1), particularly in the region of maximal CRISPRa effectiveness. We also agree that this is unlikely to be a distinct feature of CRISPRa, as there are indeed troughs of CRISPRa activity downstream of the TSS and upstream of the effective CRISPRa region, and previous in vitro results demonstrated that the nucleosomes directly impeded (d)Cas9 activity independent of any effector domain (Horlbeck et al. *eLife* 2016; Isaac et al. *eLife* 2016). Instead, we believe the difference between CRISPRi and CRISPRa accessibility patterns is due in part to the limited dynamic range in this region, as the reviewers suggested; in part to the lower expression levels of genes most sensitive to CRISPRa overexpression (Gilbert et al. Cell 2014; Konermann et al. Nature 2015), resulting in more poorly phased nucleosomes; and in part to the smaller CRISPRa dataset that reduces our resolution in fitting the TSS positioning relationship. We have added a comment on this point to the main text (subsection “An integrated machine learning approach predicts highly active sgRNAs for CRISPRi and CRISPRa”, fifth paragraph).

*7) Can the authors comment on the very strong peak of effectiveness for CRISPRi sgRNAs just downstream of the TSS? This likely reflects something in addition to nucleosome-deprivation as being helpful for CRISPRi. I find the very sharp peak in maximal activity in Figure 1—figure supplement 1 to be really striking (even in comparison to previous work using less well-refined TSSs), and would encourage its inclusion in the final manuscript.*

*As the authors have noted before, CRISPRi really works best when the guide is positioned just downstream of the start site- perhaps because it is more effective to block early transcription elongation and the release from Pol2 pausing, rather than farther downstream once Pol2 is loaded with the machinery to plow through chromatin etc.?*

*For those working to block expression of novel or non-coding RNAs, we think it is worth getting this idea out there in a super-clear and obvious way- that the sweet spot for optimal guides is right downstream of the promoter.*

We agree that the region just downstream of the TSS is optimal for CRISPRi likely through a combination of nucleosome deprivation, effective KRAB domain-mediated histone methylation, and physical hindrance of early transcriptional machinery. This last hypothesis is supported by our previous tiling screens using dCas9 without an effector domain, which showed gene repression activity immediately downstream of the TSS but not in the wider window of activity seen with dCas9-KRAB (Gilbert et al. Cell 2014). That this mechanism contributes to optimal CRISPRi-mediated repression lends further support to the importance of transitioning to the more accurate FANTOM TSS annotation in designing sgRNAs for both protein-coding and non-coding genes whenever possible. We have now emphasized this finding in the main text (subsection “An integrated machine learning approach predicts highly active sgRNAs for CRISPRi and CRISPRa”, fourth paragraph).

*8) Regarding the basic sgRNA prediction algorithms developed, will these be shared upon request? We see a github site for the tool created to assess chromatin features, but not for the broader prediction platform. Could the authors indicate in the Methods how an interested user might get help with sgRNA predictions?*

We have now made the prediction algorithms and pipeline available on GitHub as well, and provide the URL in the Methods with a reference to this in the main text.